# Manganese Superoxide Dismutase as a Novel Oxidative Stress Biomarker for Predicting Paroxysmal Atrial Fibrillation

**DOI:** 10.3390/jcm11175131

**Published:** 2022-08-31

**Authors:** Hao Liu, Qiao Wang, Daiqi Liu, Ziqi Li, Yulin Fu, Gary Tse, Guangping Li, Tong Liu, Gang Xu

**Affiliations:** Tianjin Key Laboratory of Ionic-Molecular Function of Cardiovascular Disease, Department of Cardiology, Tianjin Institute of Cardiology, The Second Hospital of Tianjin Medical University, Tianjin 300211, China

**Keywords:** atrial fibrillation, inflammation, oxidative stress, biomarkers, atrial remodeling, reactive oxygen species, manganese superoxide dismutase, nicotinamide-adenine dinucleotide phosphate oxidase 4

## Abstract

Background: There is accumulating evidence indicating that inflammation and oxidative stress are involved in the pathogenesis of atrial fibrillation (AF). The role of manganese superoxide dismutase (MnSOD) in the initiation and maintenance of AF has not yet been well characterized. The aim of our study is to investigate whether or not plasma MnSOD levels are associated with AF. Methods: We enrolled a total of 130 consecutive patients with AF as the case group (paroxysmal AF: 87, persistent AF: 43) and 58 patients without a history of AF as the control group after screening. Baseline clinical characteristics, laboratory and echocardiographic parameters were collected. Plasma levels of nicotinamide-adenine dinucleotide phosphate oxidase 4 (NOX4) and MnSOD were measured by an enzyme-linked immunosorbent assay (ELISA) method. These data were compared between the different groups. The relationship between MnSOD and other parameters was assessed using Spearman correlation. Multivariable logistic regression analysis was performed to identify independent predictors of AF. The area under the curve (AUC) from receiver operating characteristics (ROC) analysis was constructed to explore the value of MnSOD in predicting the occurrence of AF. Results: The levels of MnSOD were the highest in the paroxysmal AF group, followed by the persistent AF group, and the lowest in the controls. Meanwhile, the levels in the paroxysmal AF group were significantly higher than those in the controls [322.84 (165.46, 547.61) vs. 201.83 (129.53, 301.93), *p =* 0.002], but no significant difference was found between the paroxysmal AF group and persistent AF group, as well as the persistent AF group and the controls. Spearman correlation analysis indicated that there was a significantly negative correlation between MnSOD levels and LAD (r = −0.232, *p =* 0.008) and a positive correlation between MnSOD levels and RDW-CV (r = 0.214, *p =* 0.014) in the case group. Multivariate logistic regression analysis indicated that MnSOD levels [odds ratio (OR): 1.003, 95% confidence interval (CI): 1.001–1.005, *p =* 0.002] were an independent risk factor for paroxysmal AF, and the best cut-off value of MnSOD in predicting paroxysmal AF gained by ROC curve analysis was 311.49 ug/mL (sensitivity of 52.9%, specificity of 77.6%, AUC = 0.668). Conclusion: Oxidative stress underlies the pathogenesis of AF and may play a stronger role in paroxysmal AF than persistent AF. Our study showed an independent association between increased circulating plasma MnSOD levels and the occurrence of paroxysmal AF.

## 1. Introduction

Atrial fibrillation (AF) is the most common sustained cardiac arrhythmia worldwide and is associated with substantial morbidity and mortality [1,2]. Several clinical and echocardiographic indices have been identified to predict the risk of AF occurrence. An increasing body of evidence indicates that inflammation and oxidative stress are implicated in the pathophysiology of AF. During the pro-inflammatory process, free radicals are produced, which in turn increases the local oxidative stress that may underlie the etiology of AF. Previous studies on biomarkers of inflammation and oxidative stress derived from blood in AF have shown that they are correlated with adverse clinical events, such as C-reactive protein (CRP), interleukin-2 (IL-2), interleukin-6 (IL-6), interleukin-8 (IL-8), tumor necrosis factor-α (TNF-α) and monocyte chemoattractant protein-1 (MCP-1) [3,4,5]. The oxidative damage observed during AF may contribute to the atrial remodeling process and the perpetuation of arrhythmia [4,6].

Notably, reactive oxygen species (ROS) are essential to the triggering and promotion of inflammation and oxidative stress. To counter oxidative stress caused by ROS, organisms have evolved a variety of antioxidant enzymes, mainly including superoxide dismutase (SOD), catalase (CAT) and glutathione peroxidase (GSH-Px) [7]. Excessive production of SOD appears to be implicated in atrial electrical and structural remodeling. Manganese superoxide dismutase (MnSOD), an enzyme located in mitochondria, is the key enzyme that protects mitochondria from oxidative damage [6]. However, the role of MnSOD in the occurrence and maintenance of AF has not been properly clarified. The purpose of our study was to investigate the underlying relationship between circulating plasma levels of MnSOD and AF.

## 2. Materials and Methods

### 2.1. Study Design and Population

The study protocol was approved by the Ethics Committee of the Second Hospital of Tianjin Medical University. Written informed consent was obtained from all participants. This was a prospective observational study from a single tertiary center. From January 2020 until November 2021, consecutive AF patients who were admitted to the Department of Cardiology, Second Hospital of Tianjin Medical University and subsequently underwent cryoablation for their symptomatic or drug-refractory AF were screened for enrollment in the case group. Patients without a history of AF acted as the controls, which were matched for sex, age and risk factors of coronary heart disease for AF patients. Paroxysmal AF and persistent AF were defined according to the 2020 European Society of Cardiology Guidelines: paroxysmal AF was characterized by terminating spontaneously or with intervention within 7 days of onset, while persistent AF meant continuously sustained beyond 7 days and usually required pharmacological or electrical cardioversion for termination after 7 days [2]. Another thing to note was that some patients with paroxysmal AF included in this study had definite episodes of AF recently before admission, but always maintained sinus rhythm after admission even during cryoablation, while another part not only had a history of AF, but also had intermittent AF attacks during hospitalization.

The exclusion criteria for cases were: 1. Previous history of AF catheter ablation; 2. thrombus in the left atrial appendage (LAA) detected by transesophageal echocardiography (TEE); 3. left atrial diameter (LAD) ≥ 55 mm; 4. age < 18 or >85 years old; 5. uncontrolled severe congestive heart failure [New York Heart Association functional class III-IV or left ventricular ejection fraction (LVEF) < 40%)]; 6. prior cardiac surgery; 7. history of myocardial infarction in the past 3 months; 9. severe valvular heart disease; 10. primary cardiomyopathy; 11. chronic liver or kidney dysfunction [alanine aminotransferase (ALT) or aspartate aminotransferase (AST) was more than 3 times higher than the upper limit of normal value or serum creatinine (Scr) > 2.26 mg/dL]; 12. history of stroke or transient ischemic attack (TIA) within the previous 3 months; 13. severe infectious diseases at baseline; 14. uncontrolled hyperthyroidism or hypothyroidism undergoing drug treatment; 15. obvious abnormal coagulation function and contraindications to anticoagulation; 16. patients with incomplete clinical data, failure to outpatient reexamine on time and loss of follow-up.

The control group patients were eligible for inclusion in case of suffering from mild, moderate hypertension without hypertensive emergencies/urgencies, coronary heart disease but no need for stent implantation confirmed by coronary angiography and freeing of any severe systematic disease.

### 2.2. Baseline Characteristics

Baseline clinical characteristics, laboratory and echocardiographic parameters were collected after the patients were admitted. Blood tests of ALT, AST, Scr, globulin (GLO), direct bilirubin (DBIL), total cholesterol (TC), low density lipoprotein cholesterol (LDL-c), blood urea nitrogen (BUN), white blood cell (WBC), red blood cell (RBC), hematocrit (HCT), red blood cell distribution width-coefficient of variation (RDW-CV), platelet (PLT) and platelet volume distribution width (PDW) were tested at hospital laboratory. Platelet-to-lymphocyte ratio (PLR), neutrophil-to-lymphocyte ratio (NLR) and monocyte-to-high density lipoprotein-cholesterol ratio (MHR) were calculated, as described previously [8,9,10]. To evaluate renal function, baseline creatinine clearance rate (Ccr) was calculated by the Cockcroft–Gault formula: [(140-age) × body weight (kg)] × [0.85 (if female)]/[0.818 × Scr (umol/L)], and glomerular filtration rate was estimated (eGFR) using the CKD-EPI creatinine equation: 141 × min (Scr/κ, 1)^α^ × max (Scr/κ, 1)^−1.209^ × 0.993^Age^ × 1.018 [if female] × 1.159 [if Black], κ is 0.7 for females and 0.9 for males, α is −0.329 for females and −0.411 for males, min indicates the minimum of Scr/κ or 1 and max indicates the maximum of Scr/κ or 1 [11,12].

In addition, conventional transthoracic echocardiographic (TTE) and TEE examination were performed in all patients after admission. LAD, interventricular septal thickness (IVST), left ventricular end diastolic diameter (LVEDD), left ventricular end systolic diameter (LVESD), right ventricular end diastolic diameter (RVEDD) and pulmonary artery diameter (PAD) were measured, and LVEF was acquired from apical four-chamber and two-chamber views using the modified Simpson’s biplane method of disks. TEE was implemented to rule out thrombosis in LAA prior to catheter ablation, and LAA maximum filling velocity (LAA-MFV) and emptying velocity (LAA-MEV) were also obtained by TEE. Multi-detector computed tomography three-dimensional reconstruction was carried out to expose the anatomy of pulmonary veins and left atria before ablation if necessary.

### 2.3. Assessment of Oxidative Stress Biomarkers

Blood samples were collected in case group from the sheath introduced to femoral vein preceding ablation, while for the controls fasting blood was drawn from median cubital vein after admission. Subsequently, these specimens were centrifuged uniformly at 3000 rpm for 15 min at 4 °C to obtain EDTA plasma, which was stored at −80 °C until analysis. Plasma levels of MnSOD were assayed by a commercial kit (Cusabio Biotech Co., Ltd., Wuhan, China) of enzyme-linked immunosorbent assay (ELISA) in a professional laboratory, and nicotinamide-adenine dinucleotide phosphate oxidase 4 (NOX4) levels were also measured by an ELISA method (Ousaid Biotechnology Co., Ltd., Changsha, China). The detection concentrations of MnSOD were ranged from 15.6 to 1000 pg/mL with a coefficient of variation (CV) of <8% (intra-assay) and <10% (inter-assay), and the detection range of NOX4 was 0.31–10 ng/mL with a CV of <10% (intra-assay) and <15% (inter-assay).

### 2.4. Statistical Analysis

Continuous variables that correspond to normal distribution were expressed with mean ± standard deviation or median (interquartile range) if not normally distributed. Comparisons between any two groups were tested using independent samples Student *t* test or Mann–Whitney U test, differences among multiple groups were compared by one-way analysis of variance or non-parametric Kruskal–Wallis H test, while multiple comparisons between groups were performed using Student–Newman–Keuls Q test or Kruskal–Wallis one-way analysis of variance. Categorical variables were presented as counts (percentages), and chi-square test or Fisher’s exact test was used for pairwise comparison. Correlation between MnSOD and other parameters was assessed by Spearman correlation analysis. Furthermore, multivariable logistic regression analysis was performed to identify independent predictors of AF. Adjusted odds ratio (OR) and 95% confidence interval (CI) were defined for variables that were related with each outcome. We also generated a receiver operating characteristics (ROC) curve and calculated area under the curve (AUC) to determine the best cut-off value of MnSOD in predicting the occurrence of AF with sensitivity and specificity. For statistical analysis, the software package SPSS version 26.0 (SPSS Inc., Chicago, IL, USA) was used. A *p* value < 0.05 (two-tailed) was considered statistically significant.

## 3. Results

### 3.1. Baseline Characteristics

A total of 130 patients with AF was identified, including those with paroxysmal AF (*n* = 87) and persistent AF (*n* = 43). For the control group, patients without AF matched for sex, age and risk factors of coronary heart disease were included (*n* = 58). Baseline clinical characteristics of the study population are presented in Table 1. No significant differences in age, BMI, CHA_2_DS_2_-VASc score, HAS-BLED score, history of smoking, hypertension, diabetes mellitus, stroke, hyperlipidemia, neoplasm and preoperative AF medication were observed between three groups. Compared to the control group, the proportion of females was significantly lower in the persistent AF group, and the proportion of coronary heart disease was significantly lower than that in the paroxysmal AF group. Additionally, the proportion of drinking history in the persistent AF group was significantly higher than that in the controls. Laboratory and echocardiogram parameters were summarized in Table 2. DBIL, Scr, RDW-CV and LAD in the case group were significantly higher than those in the controls. GLO, LAA-MFV and LAA-MEV in the paroxysmal AF group were significantly higher than those in the persistent AF group. Additionally, BUN, RBC, HCT, LVESD and RVEDD in the persistent AF group were significantly higher than those in the paroxysmal AF group and the controls, but LVEF was significantly lower than that in the paroxysmal AF group and the controls, and LVEDD was significantly higher than that in the controls. Other indices including ALT, AST, TC, LDL-c, Ccr, WBC, PLT, PDW, PLR, NLR, MHR, IVST and PAD were similar between three groups.

With regard to oxidative stress biomarkers, the levels of plasma NOX4 were the highest in the paroxysmal AF group, followed by the persistent AF group, and the lowest in the controls, but there was no significant difference between any two groups. As for MnSOD, the levels were also the highest in the paroxysmal AF group, followed by the persistent AF group, and the lowest in the controls, but the levels in the paroxysmal AF group were significantly higher than those in the controls [322.84 (165.46, 547.61) vs. 201.83 (129.53, 301.93), *p =* 0.002]. No significant difference was found between the paroxysmal AF group and persistent AF group, as well as the persistent AF group and the controls.

### 3.2. Multivariate Predictors of AF

Multivariate logistic regression analysis indicated that LAD (OR: 1.153, 95% CI: 1.056–1.259, *p =* 0.002), Scr (OR: 1.036, 95% CI: 1.009–1.063, *p =* 0.008), RDW-CV (OR: 2.033, 95% CI: 1.091–3.788, *p =* 0.026) and MnSOD (OR: 1.003, 95% CI: 1.001–1.005, *p =* 0.002) were independent risk factors for paroxysmal AF (Table 3). By contrast, LAD (OR: 1.749, 95% CI: 1.295–2.362, *p* <.001), LVESD (OR: 1.451, 95% CI: 1.074–1.960, *p =* 0.015) and BUN (OR: 2.652, 95% CI: 1.107–6.354, *p =* 0.029) were independent risk factors for persistent AF (Table 4).

### 3.3. Correlation between MnSOD and Other Parameters

For the AF patients, Spearman correlation analysis indicated that there was a significantly negative correlation between plasma MnSOD levels and LAD (r = −0.232, *p =* 0.008) and a positive correlation between plasma MnSOD levels and RDW-CV (r = 0.214, *p =* 0.014), but no statistically significant correlation between MnSOD and other parameters (Table 5).

### 3.4. ROC Curve of MnSOD and Paroxysmal AF

Receiver operating characteristic analysis found the optimum cut-off value of MnSOD for classifying paroxysmal AF was 311.49 ug/mL (sensitivity of 52.9%, specificity of 77.6%, AUC = 0.668) (Figure 1).

## 4. Discussion

AF has a tendency to progress over time from paroxysmal to persistent and then to permanent without effective intervention. Therefore, early identification of risk factors related to AF is of great significance for its prevention and treatment. Although the pathophysiology of AF has been extensively studied in the past few decades, the specific mechanism of the occurrence and maintenance of AF is still not completely clear so far. There is accumulating evidence to indicate that AF development and perpetuation depends on atrial remodeling, which includes two major forms: electrical and structural. The pathophysiology process of atrial remodeling is considerably complex and any long-lasting change in atrial anatomy or function can contribute to its formation [13]. Chen et al. [14] summarized the fundamental mechanisms of pathological atrial remodeling including atrial stretch, electrical remodeling, oxidative stress, atrial energy metabolism, inflammation, atrial fibrosis, fatty infiltrations and molecular signatures.

Oxidative stress refers to the cell injury or death caused by peroxidation during the process of oxidation and antioxidation, which reflects an imbalance state between the formation of oxidative damage and endogenous antioxidant protection [15]. The emergence of oxidative stress is usually accompanied by the production of various biomarkers, among them ROS such as superoxide anion (O_2_·^−^), hydrogen peroxide (H_2_O_2_), hydroxyl radicals (·OH), nitric oxide (NO), hypochlorite (HOCl) and peroxynitrite (ONOO-) which are the key to the occurrence and development of AF [15]. It has been reported that ROS are generated by a wide variety of enzymes, including nicotinamide adenine dinucleotide phosphate oxidase (NOX), xanthine oxidase (XO), nitric oxide synthase (NOS), mitochondrial enzyme, myeloperoxidase (MPO) and monoamine oxidase (MAO) [16,17]. Once cellular concentrations of ROS exceed the ability of antioxidant enzymes to clean up massive free radicals in the human body, diseases such as cancer, hypertension, diabetes, atherosclerosis, inflammation, premature aging and pulmonary fibrosis may develop [18]. Experimental and clinical studies have demonstrated that oxidative stress is implicated within the atria during AF, which may suggest a critical role in the remodeling phenomenon [19,20,21]. Furthermore, excess ROS have been closely connected in the pathogenesis of AF by impairing ion channel activity and consequent action potential initiation, repolarization or propagation [16].

Antioxidant enzymes can effectively maintain the dynamic balance of the redox process by scavenging superfluous ROS and reducing cellular oxidative stress. Superoxide dismutase (SOD) is the first and most important antioxidant enzyme family that acts against ROS, particularly O_2_·^−^, and whose major function is to catalyze the disproportionation of O_2_·^−^ to H_2_O_2_ and O_2_ [18]. Up to now, three independent SOD subtypes have been discovered from the human body, namely SOD1 (CuZn-SOD), SOD2 (Mn-SOD) and SOD3 (EC-SOD), whose genomic structure, cDNA and protein composition have been studied in detail [18]. SOD2 takes Mn^2+^ as a cofactor, also known as MnSOD, which exists in the mitochondria of aerobic cells. It is one of the first in a chain of enzymes to mediate O_2_ reduction to produce ROS, and has been proved to play a key part in promoting cell differentiation, tumor formation and preventing hyperoxia-induced pulmonary toxicity [22,23]. Li et al. [7] covered that the levels of MnSOD decreased to varying degrees in many diseases such as tumors, psoriasis, inflammatory bowel disease and neurodegenerative diseases; simultaneously, low levels of MnSOD can interrupt ROS signaling to suppress tumorigenesis, which may become a promising target for oncotherapy in the future. Moreover, studies have confirmed that this enzyme has preventive and therapeutic effects on many oxidative stress-related diseases, suggesting that developing MnSOD and its analogues as new types of anti-inflammatory drugs has potential clinical applications in the future [7].

Whilst animal experiments have confirmed that knockout of SOD2 gene in mice can cause fatal cardiomyopathy, the current research data on the role of MnSOD in various human cardiovascular diseases, including AF, are relatively limited. In 2017, Bezna et al. [24] measured the serum SOD levels of 40 young patients with cardiac arrhythmia and 40 healthy controls; the findings showed that the average value of SOD in the arrhythmia group was 61.92% lower than that in the controls and all types of arrhythmias decreased, the most obvious of which was 48.46% in paroxysmal AF. Consequently, SOD represents a valuable biomarker reflecting the relationship between oxidative stress level and cardiac arrhythmic etiology. Its remarkable reduction in AF means that the oxidative stress mechanism may be involved in the occurrence of AF.

To the best of our knowledge, this is the first study involving plasma MnSOD expression in AF patients, although MnSOD, as an antioxidant enzyme, is expected to reduce the incidence of AF. However, in contrast to the findings by Bezna et al., we identified an independent association between plasma MnSOD and the occurrence of paroxysmal AF. Specifically, we found that the levels of MnSOD were the highest in the paroxysmal AF group, followed by the persistent AF group, and then the controls. Moreover, this study indicated that patients with paroxysmal AF had significantly higher levels of MnSOD than the controls and further demonstrated that plasma MnSOD was an independent risk factor for predicting paroxysmal AF, but the causal relationship remains to be elucidated. There was no significant difference between the paroxysmal AF group and persistent AF group, as well as the persistent AF group and the controls, possibly because of the comparatively small scale of study population.

As for the reasons for the above conclusions, we speculate that there may be the following points: 1. The shorter the duration of AF, the more prominent the phenomenon of atrial oxidative stress. As a protective mechanism of antioxidation, MnSOD will increase with the enhancement of oxidative stress in order to counteract inflammation-induced oxidative stress, resulting in higher levels of MnSOD in the case group than in the controls, while the levels were higher in the paroxysmal AF group than those in the persistent AF group; 2. with the prolongation of AF duration, atrial fibrosis gradually aggravated; therefore, the MnSOD secreted by atrial myocytes will be reduced accordingly. This could also lead to the levels of MnSOD in the persistent AF group being lower than those in the paroxysmal AF group; 3. some patients with paroxysmal AF had intermittent episodes of AF after admission until cryoablation began. Rapid and irregular heart activity during each AF attack substantially increases energy outflow, which is even greater than during a persistent AF attack. Finally, MnSOD levels in the paroxysmal AF group increased significantly compared to those in the persistent AF group; 4. a decrease in MnSOD levels in patients with persistent AF in relation to paroxysmal AF suggested the activation of adaptive mechanisms or a positive result of the implemental therapies such as slowing down ventricular rate, anticoagulation, etc.; 5. the number of patients included was relatively small, which made it difficult to represent the overall situation, so there might exist a certain sampling error; 6. the blood samples of the case group were collected before each catheter ablation and the time was not fixed, while the samples of the controls were collected in the early morning fasting state, so the time was relatively fixed. Due to the different times of blood taking, the secretion rhythm of MnSOD may be different, which might affect the levels of MnSOD; 7. the history of smoking, drinking, hypertension, diabetes, coronary heart disease and tumor in the study population might be closely linked to oxidative stress, hence, the interference of these factors on the determination of MnSOD could not be excluded. Nevertheless, our study points towards the possibility that promoting the expression of MnSOD may reverse atrial remodeling and prevent AF occurrence.

Finally, prior studies have suggested that NOX-derived ROS are crucial to the formation and perpetuation of AF. Liu et al. [25] have demonstrated that there was an independent association between elevated serum NOX4 levels and paroxysmal AF and persistent/permanent AF, which supported that NOX4 participates in the pathophysiological process of AF. Although we also found that the levels of plasma NOX4 in the paroxysmal/persistent AF group were higher than those in the controls, this independent relevance was not observed in our study, which might be a result of smaller sample size and differences in patient groups.

## 5. Limitations

There are certain limitations to our study which should be realized. Firstly, this study was a single center study; owing to strict inclusion and exclusion criteria, the final selected number of patients was relatively few and might reduce the statistical power. Secondly, the basic clinical features of the study population were not completely consistent; for instance, the proportion of coronary heart disease was significantly higher in the controls than that in the paroxysmal AF group. Indeed, many previous studies have clearly shown that the oxidative stress mechanism is involved in the process of coronary atherosclerosis, so the levels of NOX4 and MnSOD are likely to increase in this kind of disease, which might affect the accuracy and representativeness of the determination results. Thirdly, despite the fact that the plasma concentrations of biomarkers are widely accepted to be stable, there are still a great deal of external and internal factors that may influence their levels, such as the recent use of antioxidant agents and the difference in blood collection time; therefore, we could not affirm whether the measured value reflected a real oxidative stress activity in the study population. Fourthly, no data were collected on the accurate AF burden, such as AF seizure frequency, duration or number of episodes. Fifthly, the prospective design of the study only showed an association, not a causation. Finally, we did not gather the information on recurrence after catheter ablation of AF, so the significance of MnSOD in terms of future AF recurrence was unavailable.

## 6. Conclusions

Oxidative stress underlies the pathogenesis of AF and may play a stronger role in paroxysmal AF than persistent AF. Our study showed an independent association between increased circulating plasma MnSOD levels and the occurrence of paroxysmal AF. MnSOD may, thus, serve as an oxidative stress biomarker. Larger prospective studies are required to expound its potential role in atrial remodeling and the underlying prognostic effects in AF.

## Figures and Tables

**Figure 1 jcm-11-05131-f001:**
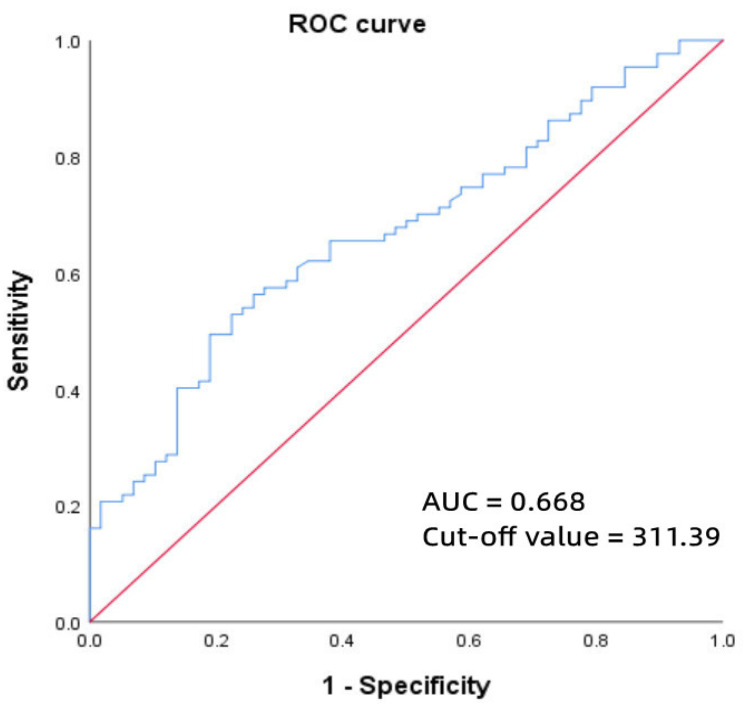
ROC curve of MnSOD for predicting paroxysmal AF. Abbreviations: ROC, receiver operating characteristic; MnSOD, manganese superoxide dismutase; AF, atrial fibrillation.

**Table 1 jcm-11-05131-t001:** Baseline clinical characteristics of study population.

	Paroxysmal AF (*n =* 87)	Persistent AF (*n =* 43)	Controls (*n =* 58)	*p* Value
Basic clinical features				
Female, *n* (%)	47 (54.0%)	14 (32.6%)	37 (63.8%)	0.007 ***
Age, years	65.8 ± 9.7	63.0 ± 8.9	62.1 ± 11.0	0.069
BMI, kg/m^2^	25.9 ± 2.9	26.7 ± 3.5	25.6 ± 3.8	0.284
CHA_2_DS_2_-VASc score	2.3 ± 1.5	2.5 ± 1.6	—	0.158
HAS-BLED score	1.5 ± 0.9	1.3 ± 0.9	—	0.271
Drinking, *n* (%)	20 (23%)	16 (37.2%)	5 (8.6%)	0.003 **
Smoking, *n* (%)	11 (12.6%)	8 (18.6%)	8 (13.8%)	0.653
Arterial hypertension, *n* (%)	59 (67.8%)	23 (53.5%)	42 (72.4%)	0.123
Diabetes mellitus, *n* (%)	21 (24.1%)	8 (18.6%)	13 (22.4%)	0.776
Coronary heart disease, *n* (%)	24 (27.6%)	14 (32.6%)	31 (53.4%)	0.005 **
Stroke, *n* (%)	16 (18.4%)	6 (14%)	4 (6.9%)	0.145
Hyperlipidemia, *n* (%)	6 (6.9%)	2 (4.7%)	6 (10.3%)	0.589
Neoplasm, *n* (%)	6 (6.9%)	5 (11.6%)	0 (0.0%)	0.504
Preoperative AF medication				
Rivaroxaban, *n* (%)	69 (79.3%)	39 (90.7%)	—	0.103
β-blockers, *n* (%)	31 (35.6%)	18 (41.9%)	—	0.491
Amiodarone, *n* (%)	38 (43.7%)	20 (46.5%)	—	0.760
Propafenone, *n* (%)	11 (12.6%)	1 (2.3%)	—	0.103
Sotalol, *n* (%)	19 (21.8%)	8 (18.6%)	—	0.669

Data were presented as means ± standard deviations or medians (interquartile range) and counts (percentages). Abbreviations: AF, atrial fibrillation; BMI, body mass index. *p* < 0.05 indicates a significant difference, ** *p*: paroxysmal AF vs. controls; *** *p*: persistent AF vs. controls.

**Table 2 jcm-11-05131-t002:** Baseline laboratory and echocardiogram parameters of study population.

	Paroxysmal AF (*n =* 87)	Persistent AF (*n =* 43)	Controls (*n* = 58)	*p* Value
Echocardiogram parameters				
IVST, mm	9.2 ± 1.6	9.3 ± 1.4	9.1 ± 1.2	0.827
LAD, mm	40.5 ± 4.3	45.4 ± 4.5	37.2 ± 5.1	<0.001 */**/***
LVEDD, mm	47.6 ± 3.8	48.5 ± 4.2	46.5 ± 4.2	0.048 ***
LVESD, mm	26.7 ± 5.3	29.7 ± 5.7	25.7 ± 3.7	<00.001 */***
RVEDD, mm	20.7 ± 2.2	21.6 ± 2.6	20.2 ± 2.1	0.010 */***
PAD, mm	22.5 ± 3.1	22.4 ± 2.9	21.4 ± 2.7	0.063
LVEF, %	62.4 ± 4.5	59.1 ± 5.0	63.5 ± 3.3	<00.001 */***
LAA-MFV, cm/s	56.3 ± 17.4	45.0 ± 16.2	—	0.001 *
LAA-MEV, cm/s	62.2 ± 22.0	40.1 ± 14.0	—	<00.001 *
Laboratory examinations				
GLO, g/L	27.9 ± 4.9	25.8 ± 4.9	26.8 ± 4.0	0.044 *
ALT, U/L	19.4 (13.9, 28.9)	19.4 (14.9, 32.0)	19.3 (14.5, 24.8)	0.672
AST, U/L	16.9 (13.2, 22.3)	18.0 (13.6, 22.2)	17.9 (15.0, 21.1)	0.505
DBIL, μmol/L	4.1 (2.8, 5.6)	4.3 (3.1, 5.9)	3.7 (2.7, 4.3)	0.029 ***
TC, mmol/L	4.9 ± 1.0	4.8 ± 0.9	4.9 ± 0.7	0.697
LDL-c, mmol/L	2.84 ± 0.86	2.94 ± 0.82	3.07 ± 0.61	0.228
BUN, mmol/L	5.9 ± 1.5	6.7 ± 1.6	5.7 ± 1.2	0.005 */***
Scr, μmol/L	72.8 ± 17.9	82.3 ± 20.0	62.6 ± 14.1	<00.001 */**/***
Ccr, mL/min	87.80 ± 25.09	89.43 ± 26.98	97.93 ± 32.95	0.096
eGFR, mL/min/1.73 m^2^	84.24 ± 15.57	81.07 ± 17.07	94.14 ± 12.54	<00.001 **/***
WBC, ×10^9^/L	6.32 ± 1.44	6.66 ± 1.60	6.52 ± 1.64	0.460
RBC, ×10^12^/L	4.52 ± 0.50	4.83 ± 0.47	4.55 ± 0.43	0.002 */***
HCT, %	41.5 ± 4.7	44.0 ± 4.7	41.3 ± 3.7	0.005 */***
RDW-CV, %	13.1 ± 1.0	13.1 ± 0.9	12.7 ± 0.5	0.007 **/***
PLT, ×10^9^/L	221 ± 56	226 ± 51	234 ± 60	0.393
PDW, %	15.9 (12.2, 16.3)	16.0 (13.7, 16.3)	16.0 (15.8, 16.3)	0.192
PLR	135.30 ± 50.52	129.82 ± 48.41	142.06 ± 51.90	0.475
NLR	2.44 ± 1.05	2.62 ± 1.07	2.49 ± 0.85	0.636
MHR	0.34 ± 0.13	0.38 ± 0.12	0.34 ± 0.15	0.336
Oxidative stress biomarkers				
NOX4, ng/mL	8.51 ± 1.59	8.17 ± 1.17	8.12 ± 1.54	0.246
MnSOD, ug/mL	322.84 (165.46, 547.61)	234.55 (149.21, 427.09)	201.83 (129.53, 301.93)	0.002 **

Abbreviations: AF, atrial fibrillation; IVST, interventricular septal thickness; LAD, left atrial diameter; LVEDD, left ventricular end diastolic diameter; LVESD, left ventricular end systolic diameter; RVEDD, right ventricular end diastolic diameter; PAD, pulmonary artery diameter; LVEF, left ventricular ejection fraction; LAA-MFV, left atrial appendage maximum filling velocity; LAA-MEV, left atrial appendage maximum emptying velocity; GLO, globulin; ALT, alanine aminotransferase; AST, aspartate aminotransferase; DBIL, direct bilirubin; TC, total cholesterol; LDL-c, low density lipoprotein-cholesterol; BUN, blood urea nitrogen; Scr, serum creatinine; Ccr, creatinine clearance rate; eGFR, estimated glomerular filtration rate; WBC, white blood cell count; RBC, red blood cell count; HCT, hematocrit; RDW-CV, red cell distribution width-coefficient of variation; PLT, platelet; PDW, platelet volume distribution width; PLR, platelet-to-lymphocyte ratio; NLR, neutrocyte-to-lymphocyte ratio; MHR, monocyte-to-high density lipoprotein-cholesterol ratio; NOX4, nicotinamide-adenine dinucleotide phosphate oxidase 4; MnSOD, manganese superoxide dismutase. * *p*: paroxysmal AF vs. persistent AF; ** *p*: paroxysmal AF vs. controls; *** *p*: persistent AF vs. controls.

**Table 3 jcm-11-05131-t003:** Multivariate logistic regression analysis on predictors of paroxysmal AF.

	β	SE	Wald	*p* Value	OR	95% CI
LAD	0.142	0.045	10.062	0.002 *	1.153	1.056–1.259
Scr	0.035	0.013	7.047	0.008 *	1.036	1.009–1.063
RDW-CV	0.709	0.318	4.988	0.026 *	2.033	1.091–3.788
MnSOD	0.003	0.001	9.762	0.002 *	1.003	1.001–1.005

Abbreviations: AF, atrial fibrillation; LAD, left atrial diameter; Scr, serum creatinine; RDW-CV, red cell distribution width-coefficient of variation; MnSOD, manganese superoxide dismutase; β, regression coefficient; SE, standard error; OR, odds ratio; CI, confidence interval. * Significant *p* value.

**Table 4 jcm-11-05131-t004:** Multivariate logistic regression analysis on predictors of persistent AF.

	β	SE	Wald	*p* Value	OR	95% CI
LAD	0.981	0.439	4.981	0.026 *	2.667	1.127–6.310
LVESD	−0.920	0.500	3.382	0.066	0.399	0.150–1.062
BUN	0.718	0.601	1.426	0.232	2.050	0.631–6.659
Scr	0.060	0.132	0.209	0.648	1.062	0.820–1.376
RDW-CV	0.080	0.084	0.905	0.341	1.083	0.919–1.277

Abbreviations: AF, atrial fibrillation; LAD, left atrial diameter; LVESD, left ventricular end systolic diameter; BUN, blood urea nitrogen; Scr, serum creatinine; RDW-CV, red cell distribution width-coefficient of variation; β, regression coefficient; SE, standard error; OR, odds ratio; CI, confidence interval. * Significant *p* value.

**Table 5 jcm-11-05131-t005:** Spearman correlation analysis of MnSOD with other parameters.

	MnSOD	
	r	*p* Value
Age	0.043	0.626
LAD	−0.232	0.008 *
LAA-MFV	−0.013	0.882
LAA-MEV	−0.077	0.385
RDW-CV	0.214	0.014 *
PLR	0.054	0.544
NLR	0.104	0.240
MHR	0.066	0.365
NOX4	−0.045	0.609

Abbreviations: MnSOD, manganese superoxide dismutase; LAD, left atrial diameter; LAA-MFV, left atrial appendage maximum filling velocity; LAA-MEV, left atrial appendage maximum emptying velocity; RDW-CV, red cell distribution width-coefficient of variation; PLR, platelet-to-lymphocyte ratio; NLR, neutrocyte-to-lymphocyte ratio; MHR, monocyte-to-high density lipoprotein-cholesterol ratio; NOX4, nicotinamide-adenine dinucleotide phosphate oxidase 4. * Significant *p* value.

## Data Availability

The data is available by contacting the corresponding authors.

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
