# Peer review of "Manganese Superoxide Dismutase as a Novel Oxidative Stress Biomarker for Predicting Paroxysmal Atrial Fibrillation"

_jcm, 2022, doi:10.3390/jcm11175131_

Round 1
Reviewer 1 Report
I read with very interest the paper "Manganese superoxide dismutase as a novel oxidative stress biomarker for predicting paroxysmal atrial fibrillation". Inflammation plays a ket role in disease and AF as well and this was the main aim of the paper. There was an increased values of MnSOD in patients with paroxysmal AF, followed by persistent AF and control. Conversely, there was no difference in NOX4 values among the aforementioned groups.
These results are in contrast with previous studies which supported low levels of MnSOD in patients with chronic disease (chronic inflammation) such as tumors, inflammatory bowel disease and so on. Another study reported high level of NOX4 in patients with AF (irrespectively is paroxysmal or persistent), whereas authors reported no difference between patients with or without AF. The results of this study are in conflict with previous studies and this may increase the interest of this article. However, the authors should discuss with more emphasis their results, highlighting the possible underlined causes of their results. Since MnSOD and NOX4 were measured from femoral vein, their values reflect the total amount of body inflammation and not only restricted to the myocardial-inflammation. The higher (and significative) prevalence of coronary disease in the control group may have altered the amount of oxidative stress biomarkers, increasing the amount of NOX 4 and MnSOD also in this group. I believe the author should highlight this point, which represent an important limit of the study which may affect results.
Although there are no differences between MnSOD values among persistent AF and the other two groups, please assess if there is a MnSOD cut off as risk of AF also in persistent AF patients. If this is possible, add persistent patients in figure 1.
Minor issues:
Line 86, please add creatinine value also as mg/dl
Please add the time when the TEE was performed (pre-admission, admission)
Assessment of oxidative stress biomarker: line 124 "preceding ablation"; line 321 "during ablation". When were they performed?
Figure 1: please add some relevant information in the figure (i.e. AUC).
Reviewer 2 Report
The aim of the study is interesting to assess the role of MnSOD values in the occurrence of FA seizures and their transition to the chronic form of FAC compared to the concentration of MnSOD in the control group.
In the presented work I see some basic problems:
Group FA paroxysmalis:
There is no information whether the patient had FA during hospitalization or only had it in the recent past, and at the time of hospitalization was SR. It seems that the occurrence of FA paroxysmalis through rapid and irregular heart activity increases energy outflow, which may result in an increase in MnSOD. Unfortunately, the authors did not address this problem in any way.
Group FA persistens:
Here I assume that all patients at the time of hospitalization have FA, although the authors did not provide this information. A decrease in the MnSOD value in patients with FA persistens in relation to FA paroxysmalis may suggest the activation of adaptive mechanisms or a positive result of the implemented therapy (B-blockers, slowing down ventricular rate, antiplatelet therapy). This group additionally had the highest creatinine level, the lowest GFR and the highest amount of RBC and the highest HTC (Diuretics, Alcohol 37.2% ?)
Control group:
It is significantly different from the study groups, it is dominated by women with coronary artery disease and hypertension who rarely consume alcohol and relatively rarely smoke cigarettes for patients with coronary artery disease. The authors did not note B-blockers in this group because of FA treatment, but I think there must have been patients in this group taking B-blockers for CAV or HA. Antiplatelet therapy was also not reported despite the fact that almost 7% of patients had symptomatic stroke.
Applications:
Work in this form only confirms the increase in the concentration of MnSOD in the group with FA paroxysmalis, unfortunately, it cannot be compared with the control group built in this way. The FA persistens group is relatively small. Statistically significant differences in EF% and LAD size probably indicate an advancement in cardiac arrhythmia. The difference in the concentration of MnSOD may be too small due to the heterogeneity of this group, or perhaps there is none.
Round 2
Reviewer 1 Report
The authors replied adequately to my comments.
Author Response
Thank you very much for your consideration and recognition of our revised manuscript!
Reviewer 2 Report
Paroxysmal AF suggests an increased oxidative stress that subsides upon transition to chronic form, but the authors' work does not in any way relate to this interesting concept.
There is still no information about the time of the FA seizure and the time of collecting the material for the study.
The control group is still significantly different from the paroxysmal end persistens FA group.
